# Plasma γ-Aminobutyric Acid (GABA) Concentrations in Lactating Holstein Cows during Thermoneutral and Heat Stress Conditions and Their Relationships with Circulating Glucose, Insulin and Progesterone Levels

**DOI:** 10.3390/vetsci11030137

**Published:** 2024-03-21

**Authors:** Alicia G. Arneson, Jacob W. Stewart, MaryKate H. Byrd, George A. Perry, Michelle L. Rhoads

**Affiliations:** 1School of Animal Sciences, Virginia Tech, 3450 Litton-Reaves Hall (0306), Blacksburg, VA 24061, USA; aga98@vt.edu (A.G.A.); jwstewa2@vt.edu (J.W.S.);; 2Texas A&M AgriLife Research, Overton, TX 75684, USA; george.perry@ag.tamu.edu

**Keywords:** γ-aminobutyric acid, dairy cow, heat stress, insulin, glucose, progesterone, milk production

## Abstract

**Simple Summary:**

During the summer months, dairy cattle exhibit reduced milk production and reproductive performance due to heat stress. Previous work found that heat-stressed dairy cattle have higher levels of blood insulin. Since insulin plays a role in metabolic and reproductive regulation, researching its regulation in heat-stressed dairy cattle may lead to an improved understanding of the mechanisms behind the reduced performance during heat stress. γ-aminobutyric acid (GABA) has been shown to play a role in regulating circulating insulin concentrations and has been implicated in reproductive function in mammals. This work aims to determine the effect of heat stress on plasma GABA levels in lactating dairy cows, and to determine what, if any, relationships exist between GABA, insulin, glucose, and progesterone in dairy cows.

**Abstract:**

Heat-stressed lactating dairy cattle exhibit unique metabolic symptoms, many of which are undoubtedly involved in heat-induced subfertility. Because of its known systemic effects, we hypothesized that γ-aminobutyric acid (GABA) participates in the regulation of insulin and progesterone during heat stress. Multiparous lactating Holstein cows (n = 6) were studied during four experimental periods: (1) thermoneutral (TN; d 1–5), (2) TN + hyperinsulinemic–hypoglycemic clamp (d 6–10), (3) heat stress (HS; d 16–20), and (4) HS + euglycemic clamp (d 21–25). Blood samples were collected once daily via coccygeal venipuncture into heparinized evacuated tubes. Analysis of GABA concentrations from all four treatment periods yielded no differences. In direct comparison to TN concentrations, plasma GABA tended to decrease during the HS period (16.57 ± 2.64 vs. 13.87 ± 2.28 ng/mL, respectively, *p* = 0.06). Both milk production and plasma insulin were moderately correlated with plasma GABA (r = 0.35, *p* < 0.01; r = −0.32, *p* < 0.01). Plasma progesterone was correlated with plasma GABA concentrations during TN but not HS periods. These results are the first to indicate that peripheral GABA could be involved in the regulation of factors known to affect production and reproduction during heat stress. More research is needed to determine its precise role(s).

## 1. Introduction

γ-Aminobutyric acid (GABA) is the primary inhibitory neurotransmitter in the central nervous system. It is not thought to cross the blood–brain barrier [1], but it is also produced by several peripheral systems in quantities comparable to central nervous system (CNS) concentrations and is therefore found in systemic circulation [2]. The two tissues that are best documented as sources of peripheral GABA are ovarian and pancreatic tissues [3,4].

In the pancreas, GABA is secreted by beta cells separately and independently from insulin and stimulates insulin production in a paracrine manner via the GABA_A_ receptor [5,6]. This effect contrasts with the known inhibitory action of GABA in the CNS and is due to the differential expression of chloride cation cotransporters in the two tissues [7]. γ-Aminobutyric acid also has an inhibitory effect on the secretion of glucagon from pancreatic alpha cells via the GABA_A_ receptor and insulin acts in a paracrine manner to upregulate the expression of these receptors on the alpha cell surface [8,9]. Glucagon works counter to insulin when conditions necessitate an increase in blood glucose by stimulating gluconeogenic pathways.

Binding sites for GABA have been documented in mammalian ovaries, in particular on mammalian granulosa cells [10]. Additionally, in preliminary work, inhibition of GABA_A_-type receptors with muscimol was shown to decrease the expression of bone morphogenetic proteins 2 and 6 in granulosa cells [11]. The effects on bone morphogenetic proteins are noteworthy because they are important for the early phases of follicular growth [12]. Inhibition of GABA_A_-type receptors with muscimol also decreased expression of the luteinizing hormone receptor in granulosa cells [11]. Peluso and Pappalardo [13] found that using bicuculline to inhibit GABA_A_-type receptors inhibited progesterone’s ability to prevent insulin-dependent mitosis, as well as apoptosis, of granulosa cells. Taken together, these findings suggest multiple roles for GABA and its receptors in mediating granulosa cell growth and follicular development.

The secretion and actions of GABA may be of particular interest for lactating dairy cows experiencing heat stress. The reproductive performance of dairy cows is known to be hindered during the summer months as a result of heat stress [14,15,16]. This is most directly observable on farms in the form of depressed conception rates. Successful establishment of pregnancy is a multi-faceted process involving numerous tissues, hormones and signaling pathways. Exposure to heat stress alters many of the processes affected by GABA, including follicular development and dynamics, the estrous cycle length, and progesterone production [17,18,19]. 

Further evidence for the involvement of GABA in the heat stress response is the fact that lactating dairy cows typically exhibit a state of hyperinsulinemia during heat stress [20]. This increase in circulating insulin is peculiar because it exists concurrently with a reduction in dry matter intake and hypoglycemia [20,21]. It is currently unknown how this altered glycemic state is established or regulated and what physiological purpose it may have. Perturbations in circulating insulin and glucose are likely involved in the disruption of reproductive processes during heat stress, as both are important for fertility. Furthermore, given its involvement in the regulation of ovarian and pancreatic processes, we hypothesized that GABA is involved in the upregulation of circulating insulin concentrations commonly observed during periods of heat stress and that GABA may be related to circulating progesterone concentrations. 

The objectives of this study were to determine the effect of heat stress on plasma GABA concentrations in multiparous lactating Holstein cows and to investigate the relationships between plasma GABA, progesterone, and glucose and insulin, alongside other physiological parameters.

## 2. Materials and Methods

### 2.1. IACUC Statement

All animal procedures performed during this experiment were approved by the Virginia Tech Institutional Animal Care and Use Committee.

### 2.2. Animals and Experimental Design

This experiment was conducted using cows from the Virginia Tech Dairy Science Complex. This work was conducted in the months of January, February and March in order to avoid any confounding effects from previous exposure to ambient heat stress. These cows represented a subset from an ongoing experiment described in Stewart et al., 2022 [22]. Six multiparous lactating Holstein cows with clinically normal periparturient periods (63.33 ± 2.35 DIM, 3.17 ± 0.40 lactations) were paired and randomly placed in one of two environmentally controlled rooms for four experimental periods: (1) thermoneutral (TN; d 1–5), (2) TN + hyperinsulinemic–hypoglycemic clamp (HHC; d 6–10), (3) HS (d 16–20), and (4) HS + euglycemic clamp (EC; d 21–25) as previously described [22]. Two transition periods were instated to allow the cows time to acclimate to the experimental conditions and husbandry procedures. The first was applied for three days before the start of the experiment (d −3 to −1). The second acclimation period allowed acclimation to heat stress and was applied for five days after the termination of the HHC period. A diagram of the treatment protocol can be found in Figure 1. The duration of the treatment periods was limited by the length of time the HHC could be applied without affecting dry matter intake [22]. Cows were milked twice daily (1230 and 0030 h), and milk yields were recorded at each milking. All cows were individually fed a TMR (Total Mixed Ration; 16.8% CP, 1.6 Mcal/kg NE_L_ on a dry matter basis) consisting of corn silage, milk cow concentrate, dry ground corn, brewers grain and whole cottonseed [22] twice daily (1230 and 0030 h) and refusals were weighed before the 1230 feeding each day. Prior to placing cows into the environmentally controlled rooms, two indwelling jugular catheters were placed in each cow on either side of the neck (four in total per cow) for infusion of insulin and dextrose, as well as for blood sampling. These catheters were flushed with heparinized saline at least once daily to maintain patency.

Rectal temperatures (°C; SureTemp Plus, Welch Allyn, Inc., Skaneateles Falls, NY, USA) and respiration rates (breaths/min) were monitored throughout the experiment. Each was measured once daily (1230 h) during the TN and HHC periods and three times daily (0030, 0900, and 1230 h) during the HS and EC periods.

### 2.3. Thermoneutral and Hyperinsulinemic–Hypoglycemic Clamp Periods

The TN and HHC periods were both performed under thermoneutral conditions for lactating dairy cattle (THI = 67.1 ± 0.26 for TN and 64.1 ± 0.36 for HHC). No other treatments were applied during the TN period. The HHC period began on day 6 of the experiment and lasted for 96 h. Hand-held blood glucose monitors were used to monitor blood glucose concentrations throughout this period (Contour Next EZ, Ascensia Diabetes Care US, Inc., Parsippany, NJ, USA). Before beginning glucose and insulin infusions, 6 baseline blood glucose measurements were recorded for each cow over a 12 h period to calculate an appropriate blood glucose baseline for each cow. 

The insulin infusate was prepared by adding purified bovine insulin (I5500, Sigma-Aldrich, Inc., St. Louis, MO, USA) dissolved in 0.01 M HCl to a saline solution containing a volume of each cow’s plasma equivalent to 1.25% of the solution. This insulin solution was infused at a rate of 0.3 µg/kg of BW/h using syringe pumps (Genie Plus, Kent Scientific, Torrington, CT, USA). A 50% dextrose solution (Nova-Tech, Inc., Grand Island, NE, USA) was simultaneously infused (Plum A+, Hospira, Lake Forest, IL, USA). The rate of glucose infusion varied according to each cow’s baseline blood glucose measurements in order to maintain a hypoglycemic state (90 ± 10% of the average of the baseline blood glucose readings for each cow). Blood glucose was measured every 5 min at the onset of the infusions until the glucose infusion rate and blood glucose measurements stabilized, at which point blood glucose was measured at intervals of every 30 min or less. For cows that did not respond to the low-dose insulin infusion, the infusion rate of insulin was slowly increased to a maximum of 0.7 µg/kg/of BW/h until blood glucose fell within the desired range. 

### 2.4. Heat Stress and Euglycemic Clamp

As previously mentioned, heat stress conditions were applied for a five-day acclimation period before formally beginning the HS period. Environmental conditions sufficient to induce heat stress in lactating dairy cows continued to be applied throughout the HS and EC periods (THI = 72.2 ± 0.36 for HS and 71.1 ± 0.36 for EC). Heat stress conditions were set to vary diurnally to mimic a summer day. The room temperature reached a maximum of 32.2 °C during the afternoon, after which it cooled slowly until it reached a minimum temperature of 21.1 °C overnight. In cases where a cow failed to acclimate acceptably to the HS treatment (multiple and repeated intervention when the rectal temperature was 40.5 °C or higher), the maximum room temperature was lowered to 29.4 °C. 

No other treatments were applied during the HS period. The EC period commenced on day 21 of the experiment, and the related infusion lasted 96 h. Six baseline blood glucose measurements for the HS period were again recorded over 12 h prior to the start of the EC period. Glucose was infused (Plum A+, Hospira, Lake Forest, IL, USA) at variable rates to return blood glucose concentrations to 100 ± 10% of the average TN baseline measurement for each cow. Blood glucose was initially measured every five minutes until infusion rates stabilized, at which point it was measured at intervals no greater than every 60 min.

### 2.5. Blood Sampling

Blood samples were collected once daily (1300 h) via coccygeal venipuncture into heparinized evacuated tubes on each day of the experimental period. Blood samples were kept cold (refrigerated) until centrifugation at 2500 RPM for 20 min. Once centrifuged, plasma for GABA concentration analyses was collected and stored at −80 °C. Plasma for insulin concentration analyses was collected and stored at −20 °C.

### 2.6. Hormone and Metabolite Analysis

Plasma hormones and metabolites were measured in all daily samples collected during TN, HHC, HS and EC. Plasma GABA concentrations were determined using a competitive inhibition ELISA (Gamma-Aminobutyric Acid (Competitive EIA) ELISA Kit-LS-F10676, LSBio, Shirley, MA, USA) validated for use in bovine samples. The inter- and intra-assay coefficients of variation were 13.9% and 6.3%, respectively. Plasma insulin concentrations were determined using a sandwich ELISA (Bovine Insulin ELISA 10-1201-01, Mercodia, Uppsala, Sweden). The inter- and intra-assay coefficients of variation were 11.1% and 5.9%, respectively. All samples and standards were run in triplicate. GraphPad Prism 9.0.0 was used to interpolate the standard curves to determine the concentrations in the individual samples. Blood glucose concentrations were determined at the time of blood collection using handheld glucometers, as previously stated (Contour Next EZ). Circulating concentrations of progesterone were analyzed via a radioimmunoassay (RIA) using the methodology described in Engel et al., 2008 [23]. Intra- and inter-assay coefficients of variation for progesterone assays were 3.5% and 8.4%, respectively.

### 2.7. Statistical Analyses 

The experiment was conducted so that each cow served as its own control. The order of treatment administration could not be randomized due to the nature of the treatments. All models were fitted as generalized linear mixed models using PROC GLIMMIX (SAS 9.4). In all models, the period and day of period were included as fixed effects, and the day of period was removed from the model when it was not significant. The interaction of the period and time was not able to be modeled due to sample size limitations. The random effects were modeled hierarchically to account for the multiple subject effects. A random intercept for each cow was included to account for dependence among all observations within a cow. The autocorrelation of observations from each cow within a period was modeled as an R-side effect with a first-order autoregressive covariance structure. This was necessary because cows served as their own controls. For analyses of plasma GABA concentrations, a GLMM with a normal distribution and a logarithmic link function was used. For analyses of plasma progesterone, a GLMM with a gamma distribution and logarithmic link function was used. The gamma distribution was selected due to the highly skewed nature of the progesterone data. For analyses of plasma insulin, rectal temperature, and respiration rate, a normal distribution and identity link function were used. This is identical to using a linear mixed model since the model assumptions were met for plasma insulin. All post hoc pairwise comparisons were adjusted using the Tukey–Kramer adjustment for multiple comparisons.

Additionally, RCORR (SAS 9.4) was used to evaluate correlational relationships between plasma GABA concentrations and the other production parameters and to evaluate the relationship between the change in plasma GABA concentrations from thermoneutral to heat stress conditions alongside the corresponding change in circulating glucose and insulin concentrations. 

Because plasma GABA concentrations in heat-stressed dairy cattle have never been compared to a thermoneutral control, two versions of the data were used for analysis. The first version included data from all four treatment periods. The second analysis excluded data from HHC and EC periods so that GABA concentrations during TN were directly compared to concentrations during HS. The results of each analysis are reported explicitly. 

## 3. Results

Respiration rates differed between the TN and HS periods as expected (TN = 58.1 ± 3.7 bpm, HS = 71.8 ± 3.8 bpm, *p* = 0.0005). Rectal temperatures also differed between the TN and HS periods as expected (TN = 38.9 ± 0.2 C, HS = 39.7 ± 0.2 C, *p* = 0.01).

Plasma insulin differed by period both in the analyses with and without the HHC and EC periods included. In the analysis with all four periods included, plasma insulin concentrations during the HS and EC periods were greater than those during the TN period (Table 1). Recall that insulin and glucose were experimentally manipulated during the HHC, and thus, insulin tended to be elevated during HHC compared to TN (Table 2, *p* = 0.067). In the analysis including only data from the TN and HS periods, plasma insulin was increased during the HS period (Table 1).

In the analysis including all four experimental periods, no significant period differences were found among mean plasma GABA concentrations (Figure 2). In the analysis that directly compared the TN period to HS without including either clamp period, plasma GABA concentrations during HS tended to be lower during the TN period (Figure 3).

The models of daily plasma progesterone measurements by period with and without the HHC and EC periods yielded no significant differences (Table 2).

A correlational analysis was also conducted on this data set (Figure 4) and plasma GABA concentrations were moderately correlated with daily milk production (r = 0.35, *p* = 0.0002; Figure 5) and daily plasma insulin measurements (r = −0.32, *p* = 0.0009; Figure 6). None of the other variables tested (plasma progesterone, daily feed intake, average daily rectal temperature, or average daily respiration rate) were significantly correlated with daily plasma GABA concentrations (Figure 4). As expected for heat-stressed dairy cattle, average daily rectal temperatures and respiration rates were negatively correlated with feed intake (r = −0.48, *p* < 0.0001 and r = −0.36, *p* = 0.0002, respectively; Figure 4). Plasma progesterone was correlated with the daily feed intake (r = −0.41, *p* < 0.0001), average daily respiration rate (r = 0.24, *p* = 0.0141), and average daily rectal temperature (r = 0.40, *p* < 0.0001) (Figure 4). A separate correlational analysis of the available blood glucose measurements with the corresponding plasma GABA measurements at the end of the TN and HS periods yielded a strong negative correlation (r = −0.63, *p* = 0.0273; Figure 7). It is worth noting that the sample size available for this analysis was small, however. 

Correlational analyses were also conducted separately for the TN and HS periods to identify changes in relationships between the physiological variables. In this analysis, it was found that the correlation between plasma GABA and daily milk production was greater during HS than TN periods (TN: r = 0.333, *p* = 0.08; HS: r = 0.583, *p* = 0.003). Interestingly, plasma progesterone was moderately correlated with plasma GABA during TN, but not during HS periods (TN: r = 0.453, *p* = 0.01367; HS: r = −0.280, *p* = 0.1858). The full results of this analysis can be found in the Appendix A.

## 4. Discussion

Undeniably, reproductive- and lactation-based responses to heat stress in cattle are quite complex, and there is no one solution capable of improving productivity and fertility in heat-stressed cattle. For these reasons, it is necessary to develop a complete understanding of the physiological adaptations that occur when cattle are exposed to heat stress. The hyperinsulinemic–hypoglycemic state that is typically observed in heat-stressed cattle is one such adaptation that requires a deeper understanding. In particular, the increase in circulating insulin concentrations is counterintuitive, as cattle exposed to heat stress reduce their dry matter intake [20,21]. Unfortunately, the mechanism(s) driving the increase in circulating insulin concentrations have not been elucidated. 

The HS treatment applied during this experiment successfully elicited increases in rectal temperatures and respiration rates, which were comparable to those observed in previous heat stress experiments [20,21]. As expected, circulating insulin concentrations in the current experiment were greater during the HS period than during the TN period. Insulin is known to have positive effects on postpartum follicular development [24,25], but damages oocyte quality. Diets designed to increase circulating insulin concentrations in cows and heifers decrease in vitro blastocyst development [26,27,28]. Furthermore, states of hyperinsulinemia are known to decrease oocyte quality in cattle, mice and humans [26,29,30,31]. These data suggest that insulin has opposing effects on two different aspects of fertility; it has positive effects on postpartum follicular development, and negative effects on the developmental competence of oocytes. Garnsworthy and colleagues [25] exploited these opposing effects by target feeding dairy cattle for high or low circulating insulin concentrations during the postpartum period. As expected, pregnancy rates were immensely improved by administering an insulin-stimulating diet until the first postpartum ovulation and then switching cows to a lower insulin diet (high–low treatment). At 120 d postpartum, 60% of these cows were pregnant compared to only 27% of the cows on other dietary treatments (high–high, low–high or low–low diets). These results provide striking evidence for the opposing effects of insulin on follicular development and oocyte quality. 

We hypothesized GABA is one of the factors involved in the regulation of insulin during heat stress. Concentrations of GABA in the plasma of dairy cattle and the physiological consequences of changes therein are not well characterized. The majority of published studies pertaining to GABA in lactating dairy cows are based on supplementing a rumen-protected form of GABA to heat-stressed cows. Dietary supplementation of GABA in lactating dairy cows has resulted in increased feed intakes, decreased rectal temperatures, modest increases in milk production, and various changes in immune molecule concentrations [32,33]. A similar study performed under thermoneutral conditions also documented an increased feed intake and modest increases in milk production [34]. In the current study, we did not observe a significant correlation between plasma GABA and feed intake. This may imply that increased feed intakes following dietary supplementation of GABA are a pharmacological effect of exogenously increasing plasma GABA. We also cannot rule out the possibility that plasma GABA varies diurnally. If such a variation exists, however, the consistent sampling time in the current study would minimize any confounding effects of time of day. Plasma GABA was positively correlated with daily milk production in the current study, however, which is consistent with the results of the GABA supplementation studies demonstrating modestly increased milk production.

Overall, plasma GABA concentrations tended to decrease during the HS period compared to during the TN period. Knowing that insulin increases during heat stress and GABA is generally stimulatory for insulin secretion, this result was unexpected. We expected plasma GABA to increase during heat stress, which would suggest a role for GABA in the stimulation of insulin secretion during heat stress. Upon examining the GABA concentrations of individual cows, we found that four out of six cows enrolled in this experiment experienced a decrease in plasma GABA during the HS period (in comparison to TN). Interestingly, the two cows that responded with increased plasma GABA also experienced a more extreme hypoglycemic response during heat stress, based on the percent change in blood glucose. This response would be consistent with the expected increase in circulating insulin in response to greater GABA concentrations and highlights the variability in individual cow responses to heat stress. Clearly, more work is needed to further explore the relationship between endogenous circulating GABA and insulin in lactating dairy cattle. 

Although it was not the expected result, the overall reduction in plasma GABA during heat stress could be indicative of an increase in receptor populations and, thus, greater uptake by the pancreas and other GABA-responsive tissues. Previous work demonstrated that GABA_A_ receptor subunits are not found in substantial quantities on the surface of beta cells [8], which could suggest that GABA does not have much impact on the release of insulin from the pancreas at physiological concentrations. If heat stress causes an increase in receptor numbers within the pancreas, the increased sensitivity to available GABA could be a mechanism driving the increase in insulin during heat stress. In the current study, plasma insulin was weakly correlated with plasma GABA. This finding provides further evidence that plasma GABA may not be representative of GABA activity and/or concentrations within the pancreas. This makes sense if most of the GABA produced in the pancreas is used locally in an autocrine or paracrine fashion as is widely considered to be the case [35]. Unfortunately, there was no opportunity to collect pancreatic tissue for analysis of receptor populations because the cows enrolled in this study were not sacrificed. Future studies are needed for evaluation of receptor populations in the pancreas and other GABA-responsive tissues during heat stress.

Several studies have demonstrated that perfusing the pancreatic cells with GABA or GABA_A_ receptor agonists does impact the release of insulin from the pancreas [36,37,38], and Feng and co-workers [39] showed that administering GABA to mice treated with streptozotocin to induce type I diabetes largely restored insulin levels. Taking these results into account, there is reason to think that alterations in plasma GABA levels, no matter their origin, should impact insulin secretion from the pancreas. However, the correlational analysis of plasma GABA and plasma insulin presented in this work suggests that the impact may be much smaller than that of other endocrine or nervous changes yet to be investigated.

In the case that plasma GABA is directly representative of pancreatic GABA, the lack of substantial correlations between plasma insulin and plasma GABA supports the findings of Braun and co-workers [6] that the release of GABA from the pancreas is independent of the release of insulin both temporally and spatially. Some studies have implicated GABA in the insulin production process via its ability to increase the beta cell mass rather than beta cell action [40,41]. Investigating changes in beta cell mass and GABA_A_ receptor expression on the beta cell surface during heat stress in lactating cows would provide more clarification regarding whether changes in plasma GABA are indicative of an effect on insulin production.

The relationship between blood glucose and plasma GABA observed in this study is of particular interest going forward. While the precise relationship between GABA and pancreatic beta cells is somewhat unclear, there is substantial unified evidence that GABA_A_ receptors exist on the surface of alpha cells and allow GABA to mediate an inhibitory effect on glucagon release [8,9,42]. Additionally, Xu and co-workers [9] provided evidence that insulin induces GABA_A_ receptor translocation to the alpha cell membrane and that this is the mechanism by which insulin can suppress glucagon release. Therefore, the state of heat-stress-induced hyperinsulinemia may increase the GABA_A_ receptor density on the alpha cell surface. This mechanism would also explain why the two cows with the greatest decrease in plasma GABA during heat stress were also those cows who experienced increased plasma glucose during heat stress, since the reduced GABA concentrations could be related to increased utilization rather than decreases in production.

Interactions between GABA and progesterone are bidirectional and highly dependent upon the tissues and systems of interest. These interactions have been implicated in normal reproductive processes such as regulation of LH secretion [43,44] and sperm hyperactivation [45], as well as a number of maladies, including premenstrual dysphoric disorder, peripartum depression, catamenial epilepsy and polycystic ovary syndrome [46,47,48,49]. While the effects of GABA on reproductive processes have been best defined within the central nervous system, some direct actions on female reproductive tissues have been described as well. As previously mentioned, GABA and its binding sites are found in substantial quantities within the ovaries [10]. In vitro treatment of luteal cells with GABA has yielded mixed results, with some finding no effect [50] and others finding that GABA reduces luteal cell progesterone production [51]. Perhaps more importantly, progesterone and GABA receptor agonists are thought to work cooperatively to enhance the oviductal cilia activity that is necessary for gamete transport [52]. In the current study, plasma progesterone was positively correlated with plasma GABA in thermoneutral conditions, which is consistent with cooperative effects to promote fertility. During heat stress, however, no correlation existed, indicating a disruption in any concerted actions. Unfortunately, the current study was not designed in a manner that would allow deeper investigation of the relationship between plasma GABA and progesterone, and the results are limited by the unknown luteal status of the cows. Even though daily plasma GABA concentrations were not related to feed intake, they were correlated with daily milk production. This correlation was especially pronounced during heat stress. Limited research is available regarding the action of GABA within mammary tissue in mammals and none exists that is specific to bovines of any kind. The available research predominantly pertains to the role of GABA in breast cancer metastasis [53,54] or to the regulation of prolactin secretion by GABA in the hypothalamus and pituitary gland [55,56]. Because GABA does not cross the blood–brain barrier, more research is necessary to determine if GABA is produced within mammary tissues and to understand if it is capable of directly affecting the bovine mammary gland or if it acts elsewhere to influence milk production. 

## 5. Conclusions

The results of this experiment provide initial information regarding relationships between plasma GABA and pertinent physiological parameters in heat-stressed dairy cows. Additionally, this study was the first to determine how plasma GABA changes during heat stress compared to a thermoneutral control and demonstrates a tendency for plasma GABA to be reduced during heat stress. Taken together, the results provide compelling support for the further investigation of the role of GABA in heat stress responses in lactating dairy cows regarding the altered glycemic state, fertility and overall productivity during heat stress. 

## Figures and Tables

**Figure 1 vetsci-11-00137-f001:**
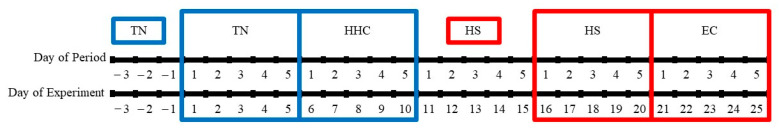
Visual representation of the experimental design. Cows (n = 6) were subjected to four experimental periods: (1) thermoneutral (TN; d 1–5), (2) TN + hyperinsulinemic–hypoglycemic clamp (HHC; d 6–10), (3) HS (d 16–20), and (4) HS + euglycemic clamp (EC; d 21–25). Cows were acclimated to tie stall housing and experimental procedures from d −3 to −1. Days 11 to 15 of the experiment served as a period for acclimation to HS.

**Figure 2 vetsci-11-00137-f002:**
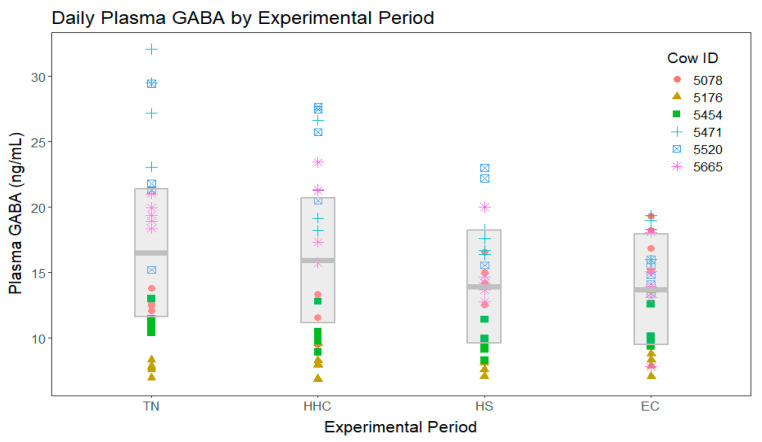
Plasma γ-aminobutyric acid (GABA) concentrations for all four experimental periods (TN = thermoneutral, HHC = hyperinsulinemic–hypoglycemic clamp, HS = heat stress, EC = euglycemic clamp). Different point styles and colors represent different individual cows, as represented in the figure legend. The gray box represents the model-estimated means (center line) for each group and their associated 95% confidence intervals. There were no significant differences detected when all four periods were included in the analysis (*p* = 0.11).

**Figure 3 vetsci-11-00137-f003:**
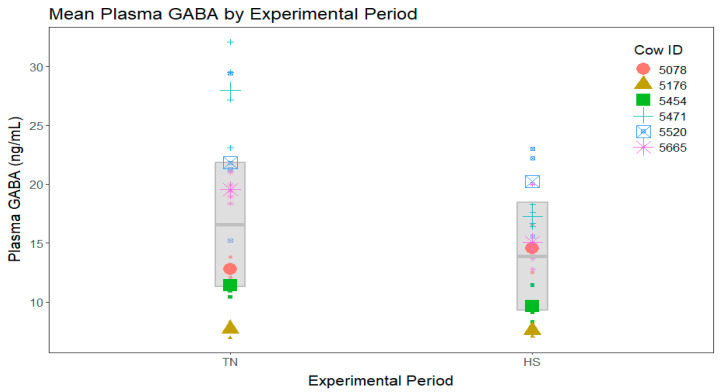
Plasma γ-aminobutyric acid (GABA) concentrations by period for the thermoneutral (TN) and heat stress (HS) periods only. Points of different shapes and colors represent different cows. Small points show the raw data. Large points show the raw mean values for each cow. The gray box represents the model-estimated means (center line) for each group and their associated 95% confidence intervals. Plasma GABA during HS tended to be reduced compared to the TN period (*p* = 0.06).

**Figure 4 vetsci-11-00137-f004:**
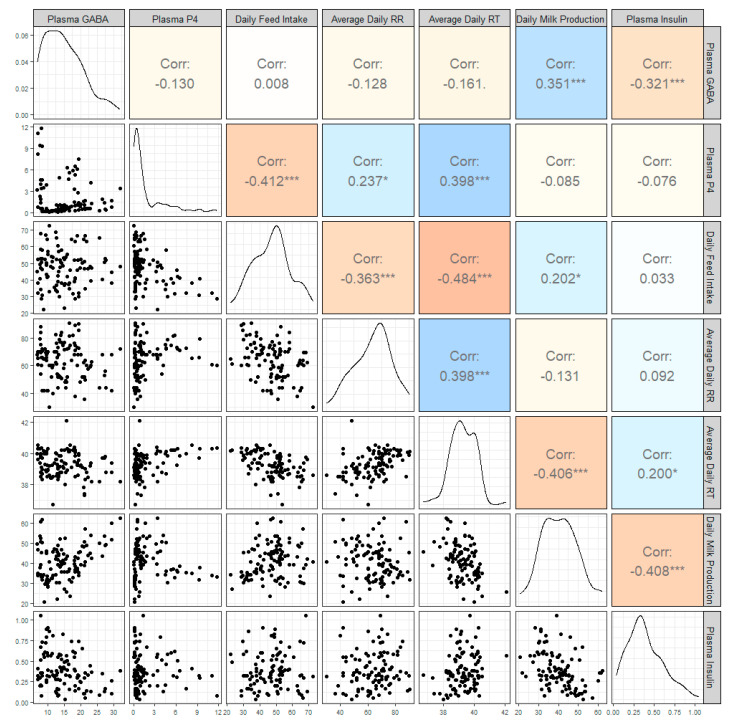
Correlogram showing the correlations between plasma γ-aminobutyric acid (GABA) concentration, plasma progesterone (P4) concentration, daily feed intake, average daily respiration rates (RR), average daily rectal temperatures (RT), and daily milk production in kilograms and the plasma insulin concentration. The bottom half of the graph shows scatterplots of each variable against the others, and the top half shows the Pearson correlation coefficient for each pairing. A correlation value marked with * shows that the *p*-value is <0.05; *** shows that the *p*-value is <0.01. The diagonal shows a density plot representative of the distribution of each variable.

**Figure 5 vetsci-11-00137-f005:**
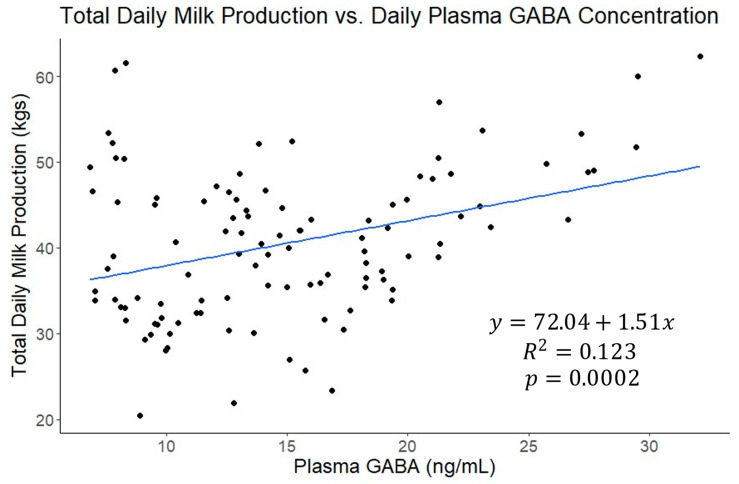
The linear fit of total daily milk production in kilograms versus plasma γ-aminobutyric acid (GABA) concentration from all four experimental periods. Plasma GABA was a significant predictor of milk production, but not sufficient by itself to explain a substantial portion of the variability in milk production.

**Figure 6 vetsci-11-00137-f006:**
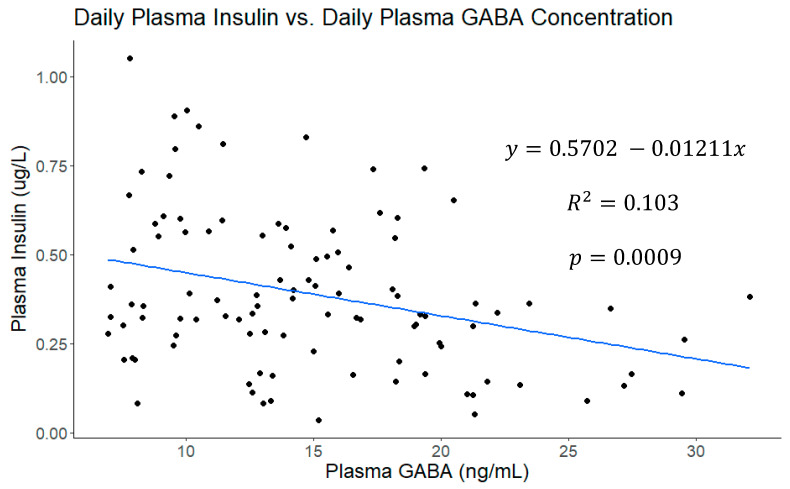
The linear fit of daily plasma insulin concentrations versus daily plasma γ-aminobutyric acid (GABA) concentrations including data points from all four experimental periods. The two are poorly correlated and plasma GABA is not explanatory for a substantial portion of variability in plasma insulin concentrations.

**Figure 7 vetsci-11-00137-f007:**
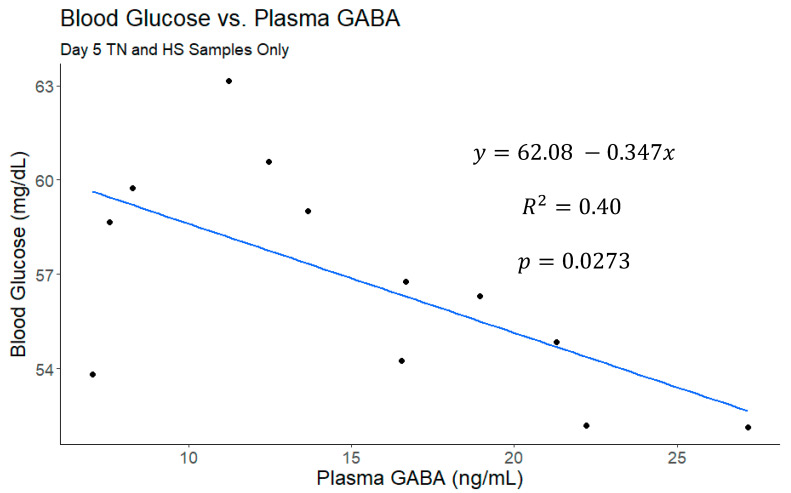
The linear fit of blood glucose versus plasma γ-aminobutyric acid (GABA) from day 5 of the thermoneutral (TN) and heat stress (HS) periods. No other samples were included because glucose measurements were only collected concurrently with GABA measurements on these days. This comparison yielded a moderate correlation, and the model including baseline GABA was significantly better than the intercept-only model.

**Table 1 vetsci-11-00137-t001:** LS means for plasma insulin by period. The results from both the analyses including all periods (TN = thermoneutral, HHC = hyperinsulinemic–hypoglycemic clamp, HS = heat stress, EC = euglycemic clamp) and including only TN and HS periods are shown. Different numbers next to the names of the periods designate significant adjusted pairwise differences between the treatments.

Means from Analysis with All Periods			
	Plasma Insulin (µg/L)	Standard Error	*p*-value
TN ^1^	0.27	0.05	0.0131
HHC ^1,2^	0.42	0.06	
HS ^2^	0.43	0.06	
EC ^2^	0.43	0.06	
**Means from analysis with all TN and HS periods only**			
	Plasma Insulin (µg/L)	Standard Error	*p*-value
TN ^1^	0.27	0.06	0.0026
HS ^2^	0.43	0.06	

**Table 2 vetsci-11-00137-t002:** LS means for plasma progesterone by period. The results from both the analyses including all periods (TN = thermoneutral, HHC = hyperinsulinemic–hypoglycemic clamp, HS = heat stress, EC = euglycemic clamp) and including only TN and HS periods are shown.

Means from Analysis with All Periods			
	Plasma Progesterone (ng/mL)	Standard Error	*p*-value
TN	0.78	0.44	0.1151
HHC	0.76	0.43	
HS	1.05	0.60	
EC	1.17	0.66	
**Means from analysis with all TN and HS periods only**			
	Plasma Progesterone (ng/mL)	Standard Error	*p*-value
TN	0.73	0.40	0.2336
HS	1.11	0.62	

## Data Availability

The data presented in this study are available on request from the corresponding author.

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
