# Peer review of "Plasma γ-Aminobutyric Acid (GABA) Concentrations in Lactating Holstein Cows during Thermoneutral and Heat Stress Conditions and Their Relationships with Circulating Glucose, Insulin and Progesterone Levels"

_vetsci, 2024, doi:10.3390/vetsci11030137_

Round 1

Reviewer 1 Report

Comments and Suggestions for Authors

The study aims to investigate the impact of plasma γ-Aminobutyric acid on the reproductive performance of cattle during heat stress, holding certain research significance. To enhance the manuscript, several modifications are suggested:

1. It is recommended to enrich the text with additional details beyond referring to the previous publication. At least to incorporate critical factors such as the experimental season/environment, which play a pivotal role in the study.

2. The term "two replicates" lacks clarity. Are you implying the division of the six cows into two groups, conducting the same experiment simultaneously? Whether having two replicates holds statistical significance?

3. Consider providing a figure for better visualization of Table 1.

4. Figure 3 differs from Figure 2 primarily in the average value, potentially lacking significant meaning. Consider consolidating key information into a single graph.

5. Although some data did not exhibit significant changes, it is advisable to include the results in the text, presenting mean ± SD/SEM, in lines 251 to 264.

6. In Figure 4, the diagonal line displays data distribution. Were there any corresponding transformations applied to the data for GABA, P4 and other indicators before analysis?

7. In Figures 5 and 6, the linear models exhibit a very small R² value. Have you considered employing nonlinear models to better elucidate the relationships between numerical values?

8. The monitored indicators are limited, is the scope of the research question too broad? Consider refining the focus, or maybe changing to a small title.

Author Response

  1. It is recommended to enrich the text with additional details beyond referring to the previous publication. At least to incorporate critical factors such as the experimental season/environment, which play a pivotal role in the study.

AU: Additional information has been added as requested. We added everything that we felt is pertinent to the study described in this manuscript, but would be happy to add more if the reviewer would like. Just let us know what additional details are desired. The work was conducted during January, February and March so that ambient outdoor conditions were thermoneutral. Heat stress was applied in environmentally controlled chambers. 

  1. The term "two replicates" lacks clarity. Are you implying the division of the six cows into two groups, conducting the same experiment simultaneously? Whether having two replicates holds statistical significance?

AU: Yes, this does mean that cows were divided into two groups as the facility could not accommodate 6 cows at once. The replicates were conducted sequentially (one immediately after the other). This had no bearing on the statistical analyses, and since the reviewer found it confusing, we deleted this to improve clarity.  

  1. Consider providing a figure for better visualization of Table 1.

AU: Thank you for this suggestion. We would prefer to keep the presentation of this data in the form of a table. Since we are presenting the means over all 4 treatment periods and also between TN and HS, we felt the figures were a little confusing. If the reviewer feels this is a critical update, however, we will be happy to oblige. 

  1. Figure 3 differs from Figure 2 primarily in the average value, potentially lacking significant meaning. Consider consolidating key information into a single graph.

AU: Thank you for this suggestion. We completely understand the reviewers point here, and debated how to approach presentation of this information. For the sake of completeness and clarity of presentation, we would prefer to keep both figures. If the reviewer feels strongly that this information needs to be consolidated, we will do so. 

  1. Although some data did not exhibit significant changes, it is advisable to include the results in the text, presenting mean ± SD/SEM, in lines 251 to 264.

AU: The analyses described in this section are correlation analyses, so the only applicable statistics to present are the correlation coefficient and p-value which are reported in the text. Additionally, figure 4 shows the results for all of the analyses, even those that were not significant.  If we have misunderstood the request, please let us know.

  1. In Figure 4, the diagonal line displays data distribution. Were there any corresponding transformations applied to the data for GABA, P4 and other indicators before analysis?

AU: No transformations were applied to any of the variables prior to these analyses. 

  1. In Figures 5 and 6, the linear models exhibit a very small R² value. Have you considered employing nonlinear models to better elucidate the relationships between numerical values?

AU: It is not clear what is meant by “non-linear”. If the reviewer is referring to the addition of polynomial terms to add curvature to the relationship, we did consider this option. However there is only sparse data available at the upper end of the range for plasma GABA so the seeming curvature in the relationship is likely spurious. Additionally, the goal of these figures and statistics is purely exploratory with the hope that others interested in the topic might design studies around interesting findings that are better suited to evaluate more complex relationships between these variables. 

  1. The monitored indicators are limited, is the scope of the research question too broad? Consider refining the focus, or maybe changing to a small title.

AU: The reviewer makes a good point. The title has been revised to more accurately reflect the scope of the work. 

Reviewer 2 Report

Comments and Suggestions for Authors

Material and methods

Paragraph 2.2

Was production in the previous lactation also taken into account when choosing the animals?

Information relating to the diet administered and the distance from birth in which the animals were fed is missing (important data to evaluate whether the animals chosen were homogeneous).

indicate the instrument used to measure rectal temperature

Paragraph 2.5

Check the time of the blood sample which, if confirmed, is only 1 hour from the administration of the meal and therefore the effect on insulin variations must be taken into consideration

Results

Discuss figure 4 in more detail

Author Response

Material and methods 

Paragraph 2.2 

Was production in the previous lactation also taken into account when choosing the animals? 

AU: Thank you for the question, but no, production from the previous lactation was not taken into account when animals were chosen. However, since all animals were subjected to all experimental periods, each animal was able to serve as their own control (i.e., there were no concerns about production differences between animals assigned to treatment vs control because all cows went through all control and treatment periods).  

Information relating to the diet administered and the distance from birth in which the animals were fed is missing (important data to evaluate whether the animals chosen were homogeneous). 

AU: Diet ingredients, crude protein % and NEL have been added as suggested. By distance from birth, is the reviewer referring to days in milk? If so, that information can be found in 2.2, and the cows were 63.33±2.35 DIM.  

indicate the instrument used to measure rectal temperature 

AU:  Added as suggested.

Paragraph 2.5 

Check the time of the blood sample which, if confirmed, is only 1 hour from the administration of the meal and therefore the effect on insulin variations must be taken into consideration 

AU: The authors understand the reviewer’s concern. To minimize effects of variation in all metabolites and metabolic hormones, the blood samples were taken consistently at the same time each day. Therefore, relative comparisons are valid without further accounting for the meal-induced changes.  

Results 

Discuss figure 4 in more detail 

AU: We have thoroughly reviewed our discussion of figure 4, and we are unsure how we could expand this discussion further. Could the reviewer please provide more direction on what details need to be covered in more depth? 

Reviewer 3 Report

Comments and Suggestions for Authors

I found the paper very interesting due to the novelty of the subject and investigations. My main concern regards the experimental protocol design:

- which were the criteria for establishing duration of the phases?

-How can we be sure that the exposure to HS was sufficiently long?

-Are there indications that GABA levels are stable during the day and not fluctuating or pulsatile? otherwise only one sample during the day could not be representative

I would ask to the authors to add these information in the mat and meth section to be clearer to the reader

Author Response

I found the paper very interesting due to the novelty of the subject and investigations. My main concern regards the experimental protocol design: 

- which were the criteria for establishing duration of the phases? 

AU: The duration of the phases was based upon the maximum duration of the hyperinsulinemic-hypoglycemic clamp (HHC) procedure, meaning the maximum duration for which it was safe and would not affect feed intake. At the time this study was conducted, the low-dose HHC that was employed in this study had been administered for a maximum of 4 days (based upon previously published studies). Therefore, all treatment periods were conducted for 4 days.   

-How can we be sure that the exposure to HS was sufficiently long? 

AU: In this study, HS was applied for 5 days before any samples were collected (acclimation described in 2.2). This is actually quite lengthy compared to most studies conducted in environmental chambers, and our previous experiments indicate this is more than sufficient. This is further supported by the difference in rectal temperatures and respiration rates described in the results.  We realized we had an opportunity to further clarify the HS acclimation in Figure 1 (the experimental design).  Please see the revised figure.

-Are there indications that GABA levels are stable during the day and not fluctuating or pulsatile? otherwise only one sample during the day could not be representative 

AU: Research regarding the stability of plasma levels of GABA are sparse. Because this information was not known, we took blood samples at the same time each day to minimize the effects of changes in GABA as much as possible. There is a 1995 study in humans that showed stability in plasma GABA levels in human subjects over three years (Petty et al., 1995, Biological Psychiatry), but the authors are unaware of studies performed on shorter time scales with the necessary frequency to understand pulsatility or circadian rhythms. 

I would ask to the authors to add these information in the mat and meth section to be clearer to the reader 

AU: Thank you for these suggestions to improve the clarity of our manuscript. Information on duration of the phases has been added to the materials and methods. The rectal temperature and respiration rate differences were already presented in the first paragraph of the results, and a sentence has been added to the discussion highlighting their differences. We have also added a sentence to the discussion about the possibility of diurnal variation in plasma GABA concentrations. We feel that pulsatile changes are unlikely since we came across no evidence for such in published literature. 

Reviewer 4 Report

Comments and Suggestions for Authors

General comment

The manuscript determine that GABA plasma concentration during thermoneutral period are correlate to plasma progesterone, with milk production but not during HS period and with feed intake. These information can be interesting for a better understanding of the effect of thermal stress on the mechanism that control insulin, glucose and progesterone plasma levels.

The only concerns is that cow were at day 63.33±2.35 DIM, 3.17±0.40 lactations but no evaluation of the reproductive stage was done (type of  CL , stage of follicular development etc. etc.). Even e simple ultrasound exam of the ovary could have given these important information.

Therefore, all the information concerning fertility are correlated to the progesterone plasma levels in 6 cows without know the  reproductive stage.  Evaluation of progesterone  is not enough for a discussion on fertility.

Similarly, there are not a real determination of production parameters.

Therefore, it should be removed from the title “Related Implications for production and fertility”. And the manuscript should be rewritten considering the above considerations.

One suggestion for the title could be: Plasma γ-Aminobutyric Acid (GABA) Concentrations in Lactating Holstein Cows during Thermoneutral and Heat Stress  Conditions and its relationship with glucose, insulin and progesterone plasma levels.  

Overall, the contents of the study is  interesting, the experiment well developed, the material and methods correct and the discussion comprehensive and appropriate.  

Specific comments

LL 73- 74 please insert references.

Ll 78-79 please insert references.

L 162 Please also indicate the period (days) during which were taken the blood samples. I suppose 25 days (all experimental period) You should include this information in fig 1  

Furthermore, it should confirm the progesterone, GABA, insulin and glucose were analyzed from all the samples.

Lines 302: please insert references.

Lines 308 please insert reference.

Author Response

The manuscript determine that GABA plasma concentration during thermoneutral period are correlate to plasma progesterone, with milk production but not during HS period and with feed intake. These information can be interesting for a better understanding of the effect of thermal stress on the mechanism that control insulin, glucose and progesterone plasma levels. 

The only concerns is that cow were at day 63.33±2.35 DIM, 3.17±0.40 lactations but no evaluation of the reproductive stage was done (type of  CL , stage of follicular development etc. etc.). Even e simple ultrasound exam of the ovary could have given these important information. 

Therefore, all the information concerning fertility are correlated to the progesterone plasma levels in 6 cows without know the  reproductive stage.  Evaluation of progesterone  is not enough for a discussion on fertility. 

Similarly, there are not a real determination of production parameters. 

Therefore, it should be removed from the title “Related Implications for production and fertility”. And the manuscript should be rewritten considering the above considerations. 

One suggestion for the title could be: Plasma γ-Aminobutyric Acid (GABA) Concentrations in Lactating Holstein Cows during Thermoneutral and Heat Stress  Conditions and its relationship with glucose, insulin and progesterone plasma levels.   

AU: We appreciate the time the reviewer has put into considering our manuscript. The reviewer makes a very good point about the scope and limitations of our experiment. The title has been changed as suggested. We understand the importance of the title for introducing the topic of the work and hope that we have sufficiently addressed the reviewer's concerns.

Overall, the contents of the study is  interesting, the experiment well developed, the material and methods correct and the discussion comprehensive and appropriate.   

Specific comments 

LL 73- 74 please insert references. 

AU: The references at the end of this paragraph (17-19) are meant to serve as references for both of the last sentences of the paragraph. We will work with the editors to make sure we have placed our citations properly.

Ll 78-79 please insert references. 

AU: Reference has been added. 

L 162 Please also indicate the period (days) during which were taken the blood samples. I suppose 25 days (all experimental period) You should include this information in fig 1  

AU: Yes, measurements were made in all daily samples from TN, HHC, HS and EC. This information has been added to the materials and methods.  

Furthermore, it should confirm the progesterone, GABA, insulin and glucose were analyzed from all the samples. 

AU: This has been confirmed in the text. 

Lines 302: please insert references. 

AU: A reference was added. 

Lines 308 please insert reference. 

AU: Perhaps we are looking at the wrong sentence, but line 308 is simply stating that we did not have the opportunity to collect pancreatic tissue in this study, so this sentence is specific to this study alone. Please let us know if we have misunderstood. 

Reviewer 5 Report

Comments and Suggestions for Authors

The manuscript authored by Alicia G. Arneson et al., describes the plasma γ-aminobutyric acid (GABA) concentrations in lactating Holstein cows during thermoneutral and heat stress conditions and related implications for production and fertility. This paper aimed to investigate the impact of heat stress on plasma GABA levels in lactating dairy cows and to explore any potential relationships between GABA, insulin, glucose, and progesterone in dairy cows. The authors focus on 6 multiparous Holstein cows of the Virginia Tech Dairy Science Complex.

From a purely numerical and statistical point of view, the results show peripheral GABA could be involved in regulating aspects of production and reproduction during heat stress. However, the authors comment that “more research is needed to determine its precise role”. In this sense, it would have been interesting to consider the relationship between stress markers (e.g. Cortisol) and GABA.

Although the number of animals used was low, in general, the work presents a good introduction. The procedures are clearly described in great detail and appropriate statistical tests have been used. The results are represented graphically and numerically allowing a good understanding of the values obtained. Overall, the manuscript is well written with occasional grammar and syntax errors.

Comments on the Quality of English Language

Overall, the manuscript is well written with occasional grammar and syntax errors

Author Response

The manuscript authored by Alicia G. Arneson et al., describes the plasma γ-aminobutyric acid (GABA) concentrations in lactating Holstein cows during thermoneutral and heat stress conditions and related implications for production and fertility. This paper aimed to investigate the impact of heat stress on plasma GABA levels in lactating dairy cows and to explore any potential relationships between GABA, insulin, glucose, and progesterone in dairy cows. The authors focus on 6 multiparous Holstein cows of the Virginia Tech Dairy Science Complex. 

From a purely numerical and statistical point of view, the results show peripheral GABA could be involved in regulating aspects of production and reproduction during heat stress. However, the authors comment that “more research is needed to determine its precise role”. In this sense, it would have been interesting to consider the relationship between stress markers (e.g. Cortisol) and GABA. 

AU: The authors agree that this is an interesting area for further research and will consider measurement of stress markers for future studies. 

Although the number of animals used was low, in general, the work presents a good introduction. The procedures are clearly described in great detail and appropriate statistical tests have been used. The results are represented graphically and numerically allowing a good understanding of the values obtained. Overall, the manuscript is well written with occasional grammar and syntax errors. 

AU: The authors wish to thank the reviewer for taking time to read our manuscript and for the thoughtful comments.